# In Vitro Inhibitory Analysis of Rationally Designed siRNAs against MERS-CoV Replication in Huh7 Cells

**DOI:** 10.3390/molecules26092610

**Published:** 2021-04-29

**Authors:** Sherif Aly El-Kafrawy, Sayed Sartaj Sohrab, Zeenat Mirza, Ahmed M. Hassan, Fatima Alsaqaf, Esam Ibraheem Azhar

**Affiliations:** 1Special Infectious Agents Unit, King Fahd Medical Research Center, King Abdulaziz University, Jeddah 21589, Saudi Arabia; saelkfrawy@kau.edu.sa (S.A.E.-K.); hmody_1984@hotmail.com (A.M.H.); fha424@gmail.com (F.A.); eazhar@kau.edu.sa (E.I.A.); 2Department of Medical Laboratory Technology, Faculty of Applied Medical Sciences, King Abdulaziz University, Jeddah 21589, Saudi Arabia; zmirza1@kau.edu.sa; 3King Fahd Medical Research Center, King Abdulaziz University, Jeddah 21589, Saudi Arabia

**Keywords:** MERS-CoV, in silico prediction, designing, siRNAs, Huh7cells, Saudi Arabia

## Abstract

MERS-CoV was identified for the first time in Jeddah, Saudi Arabia in 2012 in a hospitalized patient. This virus subsequently spread to 27 countries with a total of 939 deaths and 2586 confirmed cases and now has become a serious concern globally. Camels are well known for the transmission of the virus to the human population. In this report, we have discussed the prediction, designing, and evaluation of potential siRNA targeting the ORF1ab gene for the inhibition of MERS-CoV replication. The online software, siDirect 2.0 was used to predict and design the siRNAs, their secondary structure and their target accessibility. ORF1ab gene folding was performed by RNAxs and RNAfold software. A total of twenty-one siRNAs were selected from 462 siRNAs according to their scoring and specificity. siRNAs were evaluated in vitro for their cytotoxicity and antiviral efficacy in Huh7 cell line. No significant cytotoxicity was observed for all siRNAs in Huh7 cells. The in vitro study showed the inhibition of viral replication by three siRNAs. The data generated in this study provide preliminary and encouraging information to evaluate the siRNAs separately as well as in combination against MERS-CoV replication in other cell lines. The prediction of siRNAs using online software resulted in the filtration and selection of potential siRNAs with high accuracy and strength. This computational approach resulted in three effective siRNAs that can be taken further to in vivo animal studies and can be used to develop safe and effective antiviral therapies for other prevalent disease-causing viruses.

## 1. Introduction

Human coronaviruses are well known to cause both upper and lower respiratory tract infections in humans. A new human coronavirus, isolated in June 2012, was causing respiratory illness in a 60-year-old patient who died 11 days after hospitalization in Jeddah, Saudi Arabia. The etiological agent was designated as Middle East Respiratory Syndrome Coronavirus (MERS-CoV). MERS-CoV is known as the sixth coronavirus to cause respiratory illness and has now become a serious issue for the human as well as the camel population globally. The highest number of MERS-CoV cases was reported from Saudi Arabia, 2121 cases with 788 related deaths, and the case fatality rate (CFR) peaked at 37.1% [1]. This was the first highly pathogenic coronavirus to emerge after SARS-CoV. MERS-CoV-infected individuals developed symptoms like fever, shortness of breath, and even multiorgan failure in patients with comorbidities [2,3]. The large number of pilgrims performing Hajj and Umrah in Makkah, Saudi Arabia have the potential to increase the spread of MERS-CoV infections globally. The government of Saudi Arabia has taken several control measures to avoid this spread and no cases were reported to be transmitted in the Hajj seasons since the discovery of the virus. Currently, this virus has caused 940 deaths with 2589 confirmed cases and has spread to 27 countries (https://www.ecdc.europa.eu/en/middle-east-respiratory-syndrome-coronavirus-mers-cov-situation-update accessed on 10 April 2021) including a major outbreak in the Republic of Korea in 2015 [4]. Dromedary camels were identified as the animal reservoir for the transmission of MERS-CoV to the human population [4,5,6,7,8]. Additionally, the camel workers played a possible intermediary role for the spread of virus to the human population [8], but several MERS-CoV cases were reported that had no camel contact history [9].

The MERS-CoV belongs to the lineage C of Betacoronavirus (βCoVs), of the family *Coronaviridae* which is a large group of viruses known to infect humans and other species [10]. Currently, coronaviruses have been divided into four main groups—Alpha, Beta, Gamma, and Delta coronaviruses. The coronaviruses are enveloped, single-stranded, positive-sense RNA viruses with a genome size of 25 to 32 kilobases with high genetic diversity which favors a high rate of recombination and the emergence of new viral strains with novel characteristics and an extended host range [11]. Studies on the phylogeny of lineage C betacoronaviruses suggest the evolution of MERS-CoV in camels occurred prior to that in humans and with a possible exchange of genetic elements among ancestral viruses either in bats, or within the camel genetic ‘mixing vessel’, leading to MERS-CoV emergence [12].

Currently, there are no approved vaccines or antiviral therapy available against MERS-CoV, but many have reached advanced stages of development [13,14,15]. The technological progression is reflected upon a series of genome-wide molecular screening platforms and computational biology approaches that provide novel insights to prompt response against emerging viral diseases. The RNA interference (RNAi) using short interfering RNAs (siRNAs) and micro-RNA (miRNAs) technology plays a significant role in antiviral therapy against human virus infection [16,17]. There are several classes of RNAi mediators including siRNAs to facilitate antiviral immunity in many mammals including humans [17]. The status and future directions for RNAi-based drugs has been significantly reviewed recently [18,19]. In 2018, an siRNAs-based drug known as patisiran (Onpattro; Alnylam Pharmaceuticals) was approved by USFDA for the treatment of hereditary transthyretin amyloidosis (hATTR) with polyneuropathy [19].

siRNAs are short sequences of RNA ranging from 21 to 23 base pairs associated with a 5′ phosphate group and a 3′ hydroxyl group. Currently, therapeutic siRNAs and miRNAs are found to be some of the most promising biopharmaceuticals in commercial space as oligonucleotide-based next-generation medicines [16]. Various miRNA and siRNA-based candidate drugs are being evaluated in more than 20 clinical trials and many of them have been evaluated against different viruses including HIV [20], Flock house virus (FHV) [21], Dengue virus [22], Hepatitis C virus (HCV) [23], Influenza virus [24,25,26], Hepatitis B virus (HBV) [27], Human Papillomavirus (HPV) [28], SARS coronavirus (SARS-CoV) [29], MERS-CoV [30], and SARS-CoV-2 [31,32]. The potential siRNAs can be predicted and designed in silico, using a computational approach to provide filtration of potential siRNA candidates with high accuracy, target specificity, and reduced off-target effects.

A detailed analysis of the viral RNA structure and information on RNA interactions is vital for the virus replication cycle and to allow an understanding of the structure−function relationships and can help in the prediction of the genomic regions that need to be considered during the designing and development of potential antiviral agents against several viruses such as influenza [24]. Other structure-based approaches use the ability of small molecules to target specific RNA structures or recognize and bind to RNA targets based on their secondary or tertiary structures [24,33,34]. Recently, in silico designing of siRNAs against respiratory viruses has been reported [18,35,36] including evaluation of siRNAs against MERS-CoV. We have previously performed in silico prediction alongside a pilot study for investigating the antiviral activity for 10 of the described siRNAs in a different cell culture system [30,37].

In continuation of our previous investigation on the potential use of siRNA to inhibit MERS-CoV replication, we report here the in silico prediction, designing as well as cytotoxicity and antiviral activity of MERS-CoV-siRNAs in Huh7 cell lines. The purpose of this study is to evaluate the antiviral and cytotoxic effect of in silico designed siRNA molecules against MERS-CoV in Huh7 which are well differentiated human hepatocarcinoma cell line pre-exhibited for good in vitro growth and increased viral titer of MERS-CoV [38]. Huh 7 cell lines were selected as a proof of concept to evaluate the cytotoxicity and antiviral activity of siRNAs in a human cell line in order to show their safety and efficacy for future use in preclinical and clinical trials.

## 2. Results

### 2.1. Sequence Isolation and Multiple Sequence Analysis

The full-genome sequence of MERS-CoV was retrieved from NCBI-PubMed and multiple sequence alignment was performed using BioEdit software (V.7.02). The resulting sequence alignment of the MERS-CoV isolates showed high conservation amongst all the analyzed sequences in the ORF1ab gene. Based on the high sequence homology and role in virus replication, the ORF1ab gene was identified and used as a target for siRNA prediction and designing. The multiple sequence alignment result has been presented in Figure 1. The size of MERS-CoV genome is around 25–32 kb and Figure 1 shows the high sequence similarity in the ORF1ab gene among different strains.

### 2.2. Prediction and Selection of Potential siRNAs

In silico analysis, prediction, designing and scoring of potential siRNAs targeting the ORF1ab gene of the MERS-CoV was performed using online software. Potential siRNAs with no off-target matches effect on any human mRNA sequences were filtered and selected for further evaluation. During in silico prediction, many siRNAs were found to fulfill the less favorable criteria targeting a region in the ORF1ab gene of MERS-CoV. By using strong selection criteria and a stringent strategy, we have selected and filtered out a total of twenty-one functional, off-target siRNAs that were shortlisted from four hundred and sixty-two siRNAs as per the guidelines and basic rules of filtration [18,35,36,39,40]. The designed siRNAs’ length, nucleotide content and specificity and absence of off-target effects and secondary structures in the target site were taken into consideration by the used software during the design and filtering of the siRNAs as this would influence the efficiency, precision, and functionality of siRNAs, hence leading to a better silencing outcome [35].

The list of predicted siRNAs and their characteristics are presented in Table 1. The predicted siRNAs were expected to be highly specific and potent against MERS-CoV replication. Actually, 10 of the 21 siRNAs have been previously evaluated and described [30,37], and only 4 showed an antiviral effect against MERS-CoV replication. The predicted and designed 21 siRNAs were further used to evaluate the cytotoxicity and inhibition of virus replication in Huh7 cells.

### 2.3. Target Accessibility and Secondary Structure Prediction

The target accessibility plot and secondary structure prediction were performed for each siRNA. Based on the results obtained from the analysis (RNAxs software), it was observed that each siRNA had strong target accessibility and binding at a specific position with minimum free energy and frequency of thermodynamics in the viral genome. In our study, the predicted siRNAs showed minimum free energy (MFE–kcal/mol) ranging from −26.53 to −17.65 kcal/mol and frequency of thermodynamic ensemble ranging from 83.64 to 35.22% (Table 1). These values are in the range of optimum values for efficient accessibility and binding to the target RNA (RNAxs web server http://rna.tbi.univie.ac.at accessed on 10 April 2021).

The RNA secondary structures of the MERS-CoV-orf1ab were predicted using RNAfold. As an alternative to the MFE structure, we proposed the centroid structure. The centroid secondary structure in dot-bracket notation with a minimum free energy (MFE) of −2014.10 kcal/mol and the ensemble diversity is 1566.99. The free energy of the thermodynamic ensemble is −2558.78 kcal/mol. In order to predict using RNAfold, the genomic regions were broken into fragments consisting of the siRNA elements, and the secondary structure topology was identified from the larger genome sequence. The ensemble centroid, the centroid of the largest cluster, and the best centroid are closer in base-pair distance to the structure determined by comparative sequence analysis than is the MFE structure.

The target accessibility and the local alteration of its structure may have an inhibitory effect on target recognition and siRNA binding. Therefore, it is reasonable to consider it in the design of siRNA tools. Applying knowledge about viral RNA secondary structures as additional criteria in selecting siRNA molecules seems to be a promising approach. The binding of oligonucleotides depends on the thermodynamic properties and stability of the desired sites for antiviral agents and the secondary structure plays a crucial role in developing successful antiviral strategies. RNA structural motifs with long unpaired nucleotide tracks such as internal loops and bulges, long hairpin loops and long terminal single stranded regions are favorable targets. The target stability plot indicates the binding position of siRNA at a specific location in the viral genome and indicates the efficient binding and stability of siRNAs in the genome. The secondary structure of each siRNA and centroid secondary structure of ORF1ab region are presented in Figure 2 and Figure 3. The results of the target accessibility plot for each siRNA are presented in Figure 4 and Table 1.

We analyzed the secondary structure of siRNAs in MERS-CoV-orf1ab gene using an online tool RNAfold. The software predicted a total of 21 siRNAs in the MERS-CoV-orf1ab genome. All the siRNAs were designed and downloaded in forna format; siRNAs were potentially different with specific features. Detailed information on these siRNAs, including location and other features are provided in Table 1. The stem-loop RNA structure of all siRNAs and its conservation are presented across MERS-CoV-orf1ab genome.

The target accessibility plot of each siRNA (Figure 4) was predicted and plotted by using the RNAxs software. Each plot indicated the significant binding of each siRNA at their defined positions. The target accessibility and effective binding position of each siRNA have been indicated by using the dotted line in all the figures. The binding position of each siRNA has been shown separately and designated as siRNA-1 to siRA-21.

### 2.4. siRNA Transfection and Cytotoxicity Assay

The reverse transfection method was applied, and siRNAs were delivered by using Lipofectamine 2000 (ThermoFisher Scientific, Waltham, MA, USA) in grown and 60–80% confluent Huh7 cells. The cytotoxicity of siRNAs in Huh7 cells was found to be concentration dependent. None of the evaluated siRNAs showed significant cytotoxicity to Huh7 cells and the CC_50_ for all siRNAs were found to be >100 nM (Table 2).

### 2.5. Evaluation of Virus Replication Inhibition

In vitro evaluation of the inhibitory activity of siRNAs was performed through relative quantification of the viral RNA by real-time RT-PCR in both cell lysate and cell culture supernatant using positive virus control as a reference. The inhibition of MERS-CoV replication was found to be concentration dependent for 3 of the investigated 21 siRNAs (siRNA 6,16 and 19) both in the cell lysate and in the culture supernatant. The siRNAs showed IC_50_s of 26.22, 13.87 and 11.12nM for siRNA 6,16 and 19; respectively in supernatants and, 17.56, 16.25 and 5.17nM for siRNA 6,16 and 19; respectively, in cell lysate (Table 2). The inhibition curves for the three siRNAs are shown in Figure 5a,b.

## 3. Discussion

The emergence of MERS-CoV was reported for the first time in June 2012 from Jeddah, Saudi Arabia and this virus has reportedly spread to 27 countries with 2589 confirmed cases and 940 deaths globally (https://www.ecdc.europa.eu/en/middle-east-respiratory-syndrome-coronavirus-mers-cov-situation-update accessed on 10 April 2021) [1]. The highest fatality rate was reported in the Arabian Peninsula. Several research studies have been conducted and published globally, significantly contributing to the design and development of the measures for control of virus spread and disease management. Currently, there are no approved vaccine or antivirals available, but many studies have reached advanced stages of research and development focusing on antivirals and vaccines against MERS-CoV globally [13,41].

The nucleic acid-based siRNAs/miRNAs therapeutics have emerged as an alternative antiviral therapeutic option and contributed significantly to the development of antivirals against MERS-CoV [18,30,37,40]. Currently, many siRNA/miRNA-based therapeutics are being evaluated against various diseases, including those viral-borne, and RNAi technology is being applied to silence the expression of desired/undesired genes [16,42]. The first siRNA with documented effect in humans, ALNRSV01, is a 19 bp RNA duplex with two (2′-deoxy) thymidine overhangs on both 3′ ends to prevent its nuclease degradation [17]. The replication of MERS-CoV is mediated by the ORF1ab gene while the spike protein (S) gene enables the virus attachment to the host cells and is subject to immune pressure which might lead to immune escape mutations. Multiple advanced platforms and innovative strategies are being used to develop effective vaccines and antivirals against MERS-CoV. The MERS-CoV- S gene and receptor binding domain (RBD) have been used extensively for development of MERS-CoV vaccines [13,14,15,41,43,44,45]. The ORF1ab region includes two-thirds of the coronavirus genome and encodes for nonstructural proteins [39]. The benefits of using RNAi technology in therapeutics was hampered by many obstacles, such as: off-target effects, delivery, stability, and stimulation of immune responses. The continuous research effort was successful in reducing most of the obstacles to the minimum and many siRNA-based therapeutics have reached an advanced stage of research against various diseases [46,47,48]. Recently, a few siRNAs against HCV and MERS-CoV have been evaluated and found to be very efficient to inhibit the viral replication [30,49]. Only two studies have investigated the potential use of siRNAs against MERS-CoV but they only reported the in silico design using online software but none of them have evaluated the predicted siRNA for their cytotoxicity or antiviral activity [39,40]. The in silico prediction and designing of siRNAs, provided an opportunity to filter, screen and select the potential siRNAs against MERS-CoV. The effective use of online software also helps to minimize the siRNAs with undesired properties. The use of ORF1ab gene as a potential target for the design of siRNAs has been selected and published by other researchers with no in vitro validation to investigate the efficacy and potential of the designed siRNAs [39,40].

In this study, we have predicted, designed, selected, screened, and validated the cytotoxicity and antiviral efficacy of potential siRNAs in the Huh 7 cell line. Using the computational approach and bioinformatic analysis, this resulted in several siRNAs against MERS-CoV ORF1ab gene. We have designed and evaluated the secondary structure, target accessibility plot and thermodynamic properties of all 21 siRNAs and centroid secondary structure of ORF1ab gene using online software. We have evaluated all the twenty-one siRNAs because of their better target accessibility. The predicted siRNAs were chemically synthesized and evaluated by using Lipofectamine 2000 mediated delivery in Huh 7 cells. The lower binding energy of the siRNAs indicates better interaction therefore a better chance of target inhibition. The free binding energy of our predicted siRNAs ranged from −26.53 to −17.65 kcal/mol (Table 1). This low binding energy gives a better chance for the siRNA to bind to its target RNA for increased potential of antiviral activity.

The secondary structure of siRNAs provided useful information about the locations in the viral genome and their minimum free energy (MFE) with free energy of the thermodynamic ensemble (TE). It has been observed that the MFE and TE of each siRNA varied significantly, and these variations are significantly responsible for the effective binding of each siRNA in the viral genome and their potential effects on the degradation of the viral genome with the induction of RSIC and RNAi process. Reportedly, the secondary structure of siRNA provides vital information about the features for efficient binding and significantly regulating the translation and replication processes of the viral genome [24].

In our analysis, we have designed and predicted the secondary structure of 21 siRNAs. All the siRNAs formed the secondary structure and showed the variable binding free energy and thermodynamic properties. The secondary structure of a given viral RNA can be predicted and designed rationally using multiple techniques. Many bioinformatics tools for prediction and designing have been designed, developed and are being successfully used to identify new biological insights pertaining to RNA structures [50]. siRNAs can silence an unknown number of unintended genes due to the promiscuous entry of siRNAs into endogenous miRNA machinery and the recognition of complementarity to seed regions [51]. The bioinformatics-based methods become indispensable tools for designing siRNA-based therapeutics with high efficacy and minimal off-target effects. The in silico prediction of the RNA secondary structure helps to understand the structure–function relationships of target RNA to design and identify novel bioactive therapeutic compounds, effective diagnostics, and successful antisense strategy, especially against undruggable targets. Online tools use different algorithms, search databases of different species and provide comprehensive filtering, and evaluate seed duplex Tm for reduced off-target effects. The thermodynamics approaches apply multiple algorithms, such as minimizing free energy (MFE), high and defined accuracy, and sampling-based models [52]. The binding of siRNA efficacy strongly depends on thermodynamic properties and accessibility of the 3′-end of their binding sites and significantly reflects the underlying mechanisms of siRNA function for therapeutic agents [24,53]. For the MFE structure, the ensemble centroid (reflective of the high-frequency base pairs in the structure sample) and the centroid of the largest cluster, the results are comparable with marginal overall improvements by the centroids. Although the best centroids are the best predictors, these centroids cannot be defined when a reference structure is unavailable. However, it is an appealing feature that the best centroid predictions are based on only three to four clusters, on average. The centroid for a given set of structures is the structure in the entire structure ensemble that has the minimum total base-pair distance to the structures in the set. Thus, the centroid structure can be considered as the single structure that best represents the central tendency of the set. A centroid is referred to as the ensemble centroid when the set is the entire collection of structures sampled from the ensemble [54]. Despite the potential of siRNA-based therapeutics, challenges like rapid degradation, poor cellular uptake and off-target effects remain.

The centroid secondary structure of MERS-CoV-orf1ab genome was designed by online software and showed the binding position of siRNAs at different nucleotide sequences. The centroid secondary structure showed the binding of high frequency base pairs with the siRNAs. Target accessibility plays an important role in the efficiency of an siRNA. Currently, a few programs are available for computing the accessibility using the energy model. Sfold computes the accessibilities using posterior sampling techniques. Recently, a few reports have been published discussing the importance of accessibility around the target regions of functional RNAs. By using the computational predictions and experimental validation it has been demonstrated that the secondary structure of RNA and the binding site of siRNA and their effective accessibility around the target sites influence the siRNAs’ efficacy [53,55]. Additionally, the silencing activity of structurally similar duplexes with different sequences varies significantly. The thermodynamic properties of siRNA play a key role on the stages of RISC activation and mRNA target cleavage, determining the efficiency of strand dissociation, strand selection and mRNA cleavage. Any changes in the nucleotide affecting the thermodynamic stability of the duplex could change the silencing activity of siRNA [56]. Based on experimental data it has been suggested that highly active siRNAs are likely to have lower internal stabilities than less active siRNAs [57]. An siRNA or shRNA targeted to an accessible region will not necessarily be functional if the guide strand cannot successfully assemble into the RISC. The asymmetry of siRNA duplex ends is important for RISC assembly, whereas target accessibility is important for the downstream step of target recognition in the RNAi pathway. An siRNA designed for an accessible target site will not necessarily be functional, if it does not have the favorable differential stability of siRNA duplex ends for effective assembly of the guide strand into RISC [58].

In our study, the target accessibility plot of all 21 siRNAs generated from online software showed the binding position of siRNAs in the MERS-CoV-orf1ab genome. The efficient binding of siRNAs in the viral genome reflects the potential activity in the viral genome. As it is well reported that the efficient binding of siRNAs plays an important role in the RNAi pathway, the information about the secondary structure and target accessibility of siRNAs is very important for their efficient binding to the target. The asymmetry of the siRNA duplex plays a significant role to facilitate the assembly of the RNA-induced silencing complex (RISC). The estimation of target accessibility and duplex asymmetry can improve the target knockdown level significantly by nearly 40 and 26%, respectively. The duplex asymmetry has a significant upstream effect on RISC assembly and target accessibility has a strong downstream effect on target recognition [58]. The secondary structure of viral RNA is an important regulator of virus biology because it plays an important role in virion ribonucleoproteins and virus assembly and release of infectious virus particles. The terminal (5′ and 3′) ends of viral RNA fragments are highly conserved and known as virus promotors for virus replication and termination [24]. These regions form a base pairing resulting in a double stranded structure (Panhandle Structure, Figure 2) which is finally recognized and bound by viral polymerases. The fork or corkscrew model has been proposed to explain the initiation of viral RNA transcription.

The prediction of RNA secondary structure by using bioinformatics tools and in combination with experimental data, enables the design and development of novel, efficient and highly bioactive siRNA molecules with high potency. The RNA structure-based target-site selecting method provides novel prospects for other antiviral therapeutic agents, such as small molecules. To control and silence gene expression, synthetic oligonucleotides, including antisense oligonucleotides, small interfering RNAs (siRNAs) and microRNAs (miRNAs) as well as catalytic nucleic acids, have been broadly investigated and their mechanism of action is based on selective binding to complementary sequences of target viral RNAs [59].

The cytotoxicity of each siRNA was evaluated, and the investigated siRNAs were found to have minimum cytotoxicity on the Huh7 cells with CC_50_s > 100 nM. Three of the investigated siRNAs were found to have concentration-dependent antiviral activities in both supernatants and cell lysates as shown in Figure 5a,b. Based on their IC_50_, three of the designed siRNAs exhibited better concentration-dependent viral inhibition as compared to others in both cell lysate and supernatant. The results obtained from this work encourage us to evaluate the combination of siRNAs in other multiple cell lines by using multiple siRNAs transfection and delivery methods in future studies. This will provide better and clearer understanding of the effectiveness and potential use of siRNA-based antiviral therapeutics against MERS-CoV.

In previous pilot studies [30,37], we reported the experimental evaluation of 10 of the designed siRNAs in HEK 293 and Vero cells and found that siRNAs 1 and 4 were the most effective in Vero cells while siRNAs 1, 2, 4, 6 and 9 were most effective in HEK 293 cells. In this study, siRNAs 2, 6, 16 and 19 were found to be the most effective indicating that the cell type affects the inhibitory activity of the siRNA. This change in activity with the cell type needs to be taken into account when performing the in vivo evaluation of the siRNAs. Dromedary camels are known to be the animal reservoir for MERS-CoV transmission to human. A search for a camel cell line is needed to evaluate the effect of the designed siRNAs in camels in order to further perform an in vivo evaluation of the candidate siRNAs.

## 4. Materials and Methods

### 4.1. Sequence Selection and Multiple Sequence Analysis

To perform multiple sequence analysis and identification of targets for siRNA design, filtration and selection, total thirty-four full-genome sequences of MERS-CoV from human and camel were retrieved from NCBI-GenBank. The multiple sequence alignment was performed to select the highly conserved region in the viral genome using BioEdit software (Version 7.2). The accession numbers of each of the isolates with the aligned homologous region have been presented in Figure 1. Due to lack of space and the large size of MERS-CoV genome, Figure 1 is showing only the start and end regions of the genome.

### 4.2. In Silico Prediction, Selection and Synthesis of siRNAs

The full-genome of MERS-CoV was retrieved from NCBI-GenBank and used for prediction, filtration and designing of siRNAs. The ORF1ab region was selected as the target gene for siRNAs prediction and designing. Online freely available softwares were used as an integrated bioinformatics approach to predict, design, and for the final selection of siRNAs [60,61,62,63,64,65,66]. Based on the strict selection and filtration criteria, with different scoring tools and i-SCORE, a total of 21 different siRNAs were finally selected for further study [18]. Smaller siRNAs are preferable to use for mammalian cells as longer siRNAs can induce mammalian immune response. The RNA sequences and the locations of each siRNA in the viral genome with minimum free energy are presented in Table 1. The predicted and selected siRNAs were used for custom synthesis (Integrated DNA Technologies (IDT-USA).

### 4.3. Secondary Structure and Target Accessibility Prediction

The secondary structure and target accessibility plot of each siRNA was predicted by using online software known as RNAxs tool (http://rna.tbi.univie.ac.at/cgi-bin/RNAxs/RNAxs.cgi accessed on 10 April 2021). The sequences of siRNAs were separately used for secondary structure prediction. The secondary structure of each siRNA is presented in Figure 2 and centroid secondary structure of ORF1ab gene in Figure 3. The target accessibility plot of each siRNA was performed by (http://rna.tbi.univie.ac.at, http://rna.tbi.univie.ac.at/cgi-bin/RNAWebSuite/RNAfold.cgi accessed on 10 April 2021) and is presented in Figure 4. The secondary structure of the siRNAs was visualized in forna format by using online software (http://rna.tbi.univie.ac.at/forna/documentation.html accessed on 10 April 2021). The color of the secondary structure indicated green: stems (canonical helices), red: multiloops (junctions), yellow: interior loops, blue: hairpin loops.

### 4.4. Transfection of siRNA to Huh7 Cells

The chemically synthesized siRNAs were used for Huh7 cells transfection. The cells were purchased from Creative Bioarray (Shirley, NY, USA) and grown in standard DMEM medium at 37 °C in 96-well plates with 60–80% confluency (1 × 10^4^). A reverse transfection method was used by applying the Lipofectamine 2000 (ThermoFisher Scientific, Waltham, MA, USA) as transfection reagent following the manufacturer’s instructions. All the experiments were performed in triplicate. A defined serial dilution (0.1 to 50 nM) of each siRNA was prepared from 50 μM stocks with the addition of 100 μL Opti-MEM and Lipofectamine 2000 following the 30 min incubation at room temperature. The lipid complex with siRNAs was added gently to the grown Huh7 cells and incubated at 37 °C for 24 h.

### 4.5. Cytotoxicity Assay

The cytotoxicity of each siRNA in Huh7 cells was evaluated by using the MTT ([3-(4,5-dimethylthiazol-2-yl)-2,5-diphenyltetrazolium bromide]) assay kit (Invitrogen) following the manufacturer’s instructions. After transfection of cells, the media containing transfection reagents were removed and 100 μL of fresh DMEM media was added to each well and incubated for 24 h at 37 °C and 5% CO_2_. After incubation, 10 μL of MTT (12 mM) was added to each well followed by further incubation at 37 °C for 4 h. The precipitated formazan crystals were dissolved by mixing the 10% SDS-HCL solution (100 μL/well). The cells were further incubated for 4 h at 37 °C. The absorbance was measured at 570 nm using SpectraMax i3x imaging cytometer and the mean OD value was used for calculation of cytotoxicity of each siRNA applying the standard formula.

### 4.6. Analysis of MERS-CoV Replication Inhibition in Huh7 Cells

To validate the dose dependent efficacy of the synthesized siRNAs, we performed the experimental evaluation of siRNAs transfection at variable doses (50, 25, 10, 5.0, 1.0, 0.5, 0.25, and 0.1 nm) as previously described [30,37] except Huh7 cells were used for virus inoculation and real-time RT-PCR for MERS-CoV in the cell lysate and cell culture supernatant.

### 4.7. Inoculation of MERS-CoV to Huh7 Cells

The inoculation of MERS-CoV to the siRNA transfected Huh7 cells was performed following previously published protocols [30,37]. The transfected cells were virus inoculated and cells were further grown at 37 °C in 5% CO_2_ atmosphere and cytopathic effect (CPE) in Huh7 cells was observed daily for 3 days or until full CPE was developed in the positive virus control cells. All the experiments were performed in triplicate, the triplicate wells of each concentration were pooled together to provide enough volume for subsequent RNA extraction. Each experiment included positive virus control wells with no siRNA added and negative control wells (with no virus added). The cell supernatant and lysate were isolated and after adding lysis buffer, they were further used for viral RNA isolation by using QIAamp Viral RNA Mini Kit (Qiagen, Hilden, Germany) as per manufacturer’s instructions.

### 4.8. Real-Time PCR Assay

The purified viral RNA was used to perform quantitative real-time RT-PCR for evaluation of virus replication inhibition using MERS-CoV specific primers and probes (ORF1a and ORF1b) [5]. The Ct values of real-time PCR generated variable and comparable slopes of inhibition for each siRNA at various concentrations. each experiment included the positive and negative controls of the antiviral assays and an external positive and negative control. The variation of the Ct values for positive controls were found to have less than 10% variability between runs.

## 5. Conclusions

Based on the bioinformatics analysis, prediction and design of siRNAs, the secondary structure of siRNAs, centroid secondary structure of MERS-CoV-orf1ab gene, target accessibility plot of each siRNA and low cytotoxicity of all 21 siRNAs and the concentration dependent and low IC_50_ of the three effective siRNAs in this study, the future investigation will involve the in vivo evaluation of the selected siRNAs in a preclinical in vivo trial in small animal models. The use of the computational and bioinformatics approach for siRNA design, analysis, and real-time PCR and in vitro evaluation of siRNAs activity against MERS-CoV can be effectively used as an alternative approach for screening oligonucleotide-based antiviral therapeutics. By using this technique, a novel siRNA targeting other MERS-CoV genetic regions can be predicted, designed and evaluated as oligonucleotide-based antiviral therapeutics against MERS-CoV as well as other emerging viruses, including the culprit responsible for the currently prevalent pandemic, COVID-19.

## Figures and Tables

**Figure 1 molecules-26-02610-f001:**
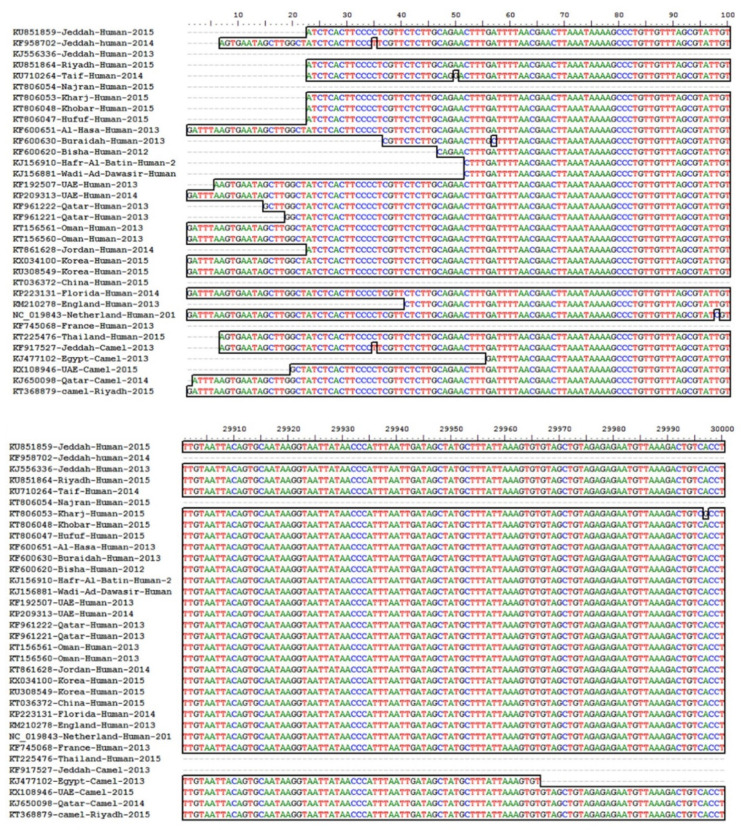
Multiple sequence alignment of selected MERS-CoVs (Nucleotide sequences start. 1……300,000 bp. Due to space limitations, we have selected the conserved regions only to display in the figure).

**Figure 2 molecules-26-02610-f002:**
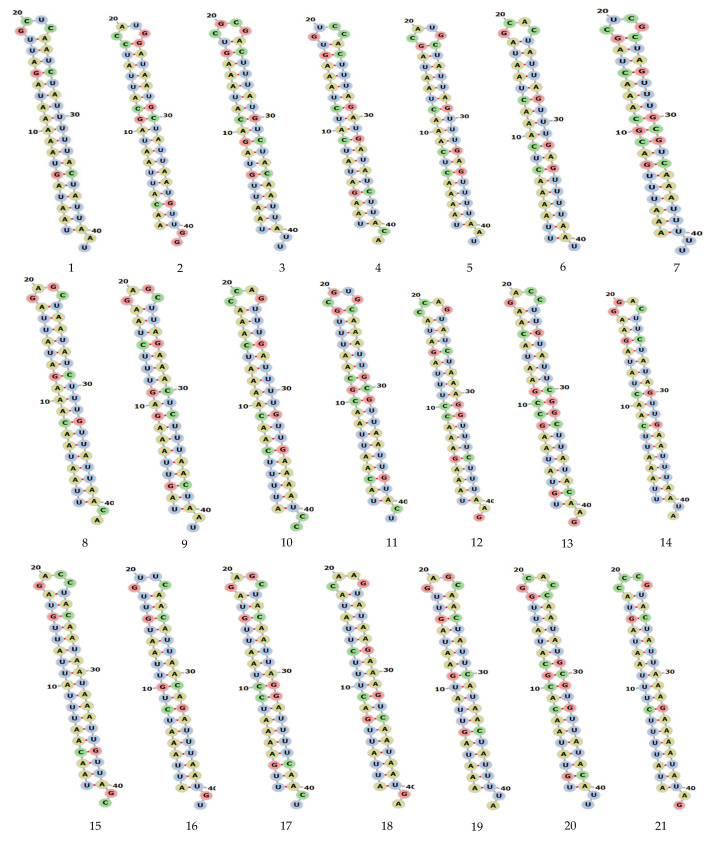
Secondary structure of siRNA MERS-CoV genome predicted by using RNAfold software and were visualized in forna Figure 3. The proposed centroid secondary structure of the MERS-CoV-ORF1ab gene with nucleotide positions was predicted using RNAfold software and downloaded in PNG format. The phosphodiester backbone is shown in orange color and the nucleotides are shown in the sticks and filled rings with elemental coloring; Green, red, yellow and blue. The color of the secondary structure indicates; green: C (canonical helices), red: G (junctions), yellow: A, blue: U hairpin loops. The structure shows all the nucleotides and their bindings with each other and forming stem, hairpin loops, canonical helices, and junctions. Chemically modified nucleotides are unpaired, in A-U or G-C pairs at helix ends, in G-U pairs anywhere, or adjacent to G-U pairs.

**Figure 3 molecules-26-02610-f003:**
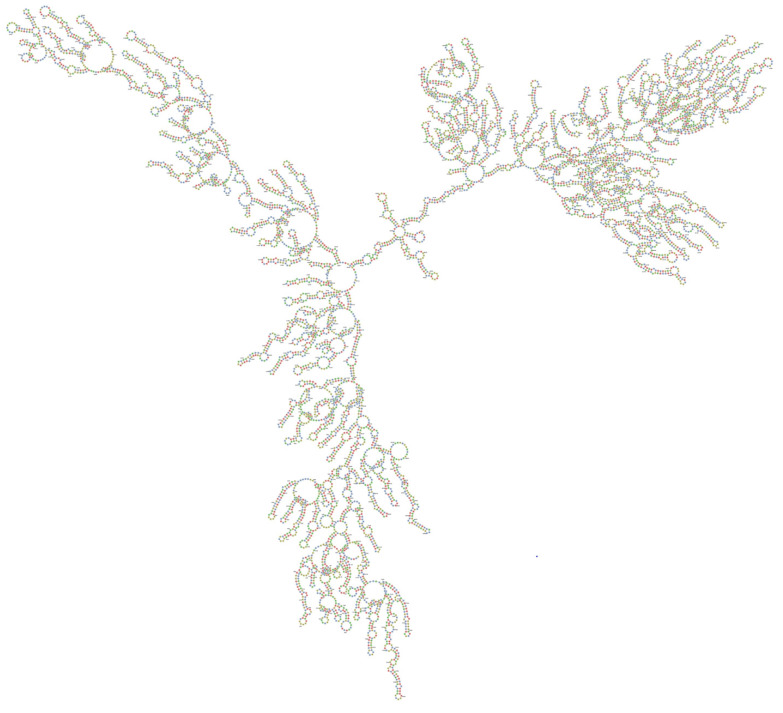
The proposed centroid secondary structure of the MERS-CoV-ORF1ab gene with nucleotide positions was predicted using RNAfold software and downloaded in PNG format. The phosphodiester backbone is shown in orange color and the nucleotides are shown in the sticks and filled rings with elemental coloring; Green, red, yellow and blue. The color of the secondary structure indicates; green: C (canonical helices), red: G (junctions), yellow: A, blue: U hairpin loops. The structure shows all the nucleotides and their bindings with each other and forming stem, hairpin loops, canonical helices, and junctions. Chemically modified nucleotides are unpaired, in A-U or G-C pairs at helix ends, in G-U pairs anywhere, or adjacent to G-U pairs.

**Figure 4 molecules-26-02610-f004:**
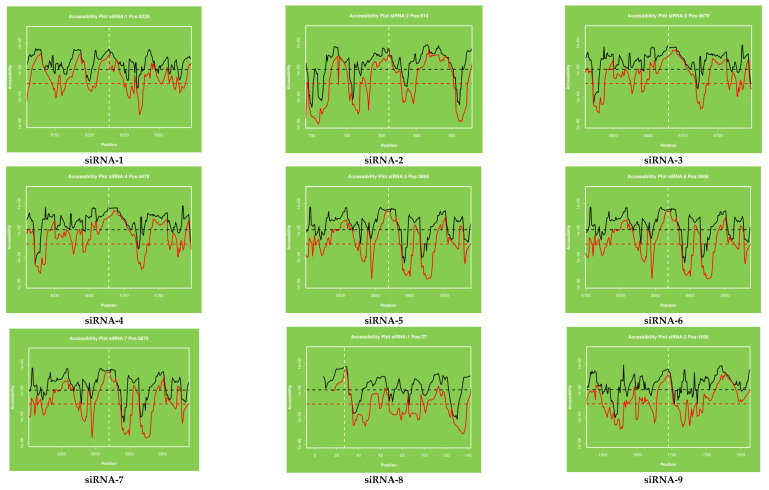
The accessibility to the target site for each nucleotide in the sequence. The proposed target accessibility plot of different siRNAs (siRNAs 1–21) was determined by RNAxs online software. Each full-length sequence target was submitted to the RNAxs online server. The target accessibility probability profile for each site targeted by the siRNA is displayed. Accessibility and targeted binding position of each siRNA have been shown as a dotted line. Each siRNA has different binding positions in the viral genome. X axis represents the efficiency and Y axis represents the binding position of siRNAs in the viral genome.

**Figure 5 molecules-26-02610-f005:**
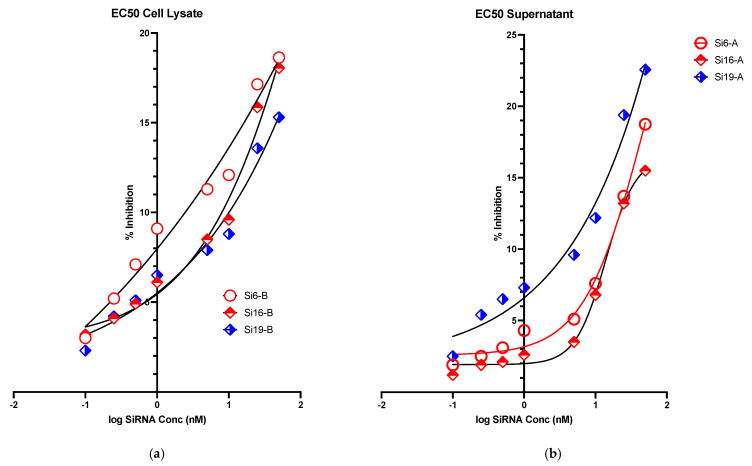
Graphical representation of Ct value of (**a**) real time-PCR result-cell lysate and (**b**) real time-PCR result-cell supernatant.

**Table 1 molecules-26-02610-t001:** List of predicted siRNAs from MERS-CoV ORF1ab gene (KF958702).

S.N.	Position of siRNA in the Genome (Start–End)	Target Sequence	Predicted RNA Oligo Sequences (5′→3′)	Minimum Free Energy (MFE- kcal/mol) and Frequency of Thermodynamic Ensemble (%)
1	791–813	agcaatctatttttactattaat	UAAUAGUAAAAAUAGAUUGCU CAAUCUAUUUUUACUAUUAAU	−17.96, 65.21
2	1615–1637	atggataatgctattaatgttgg	AACAUUAAUAGCAUUAUCCAU GGAUAAUGCUAUUAAUGUUGG	−21.80, 77.37
3	1910–1932	gcgactttatgtctacaattatt	UAAUUGUAGACAUAAAGUCGC GACUUUAUGUCUACAAUUAUU	−22.02,69.74
4	4018–4040	gacactttagatgatatcttaca	UAAGAUAUCAUCUAAAGUGUC CACUUUAGAUGAUAUCUUACA	−22.62,69.54
5	5597–5619	atgctattagtttgagttttaat	UAAAACUCAAACUAAUAGCAU GCUAUUAGUUUGAGUUUUAAU	−21.51, 83.64
6	5598–5620	tgctattagtttgagttttaata	UUAAAACUCAAACUAAUAGCA CUAUUAGUUUGAGUUUUAAUA	−19.74, 57.91
7	5819–5841	gagctagtttgcgtcaaattttt	AAAUUUGACGCAAACUAGCUC GCUAGUUUGCGUCAAAUUUUU	−24.28, 53.63
8	9495–9517	ctctaatatctttgttattaaca	UUAAUAACAAAGAUAUUAGAG CUAAUAUCUUUGUUAUUAACA	−17.97, 54.45
9	9533–9555	ctcttagaaactctttaactaat	UAGUUAAAGAGUUUCUAAGAG CUUAGAAACUCUUUAACUAAU	−22.37, 64.54
10	13,605–13,627	tggtttgattttgttgaaaatcc	AUUUUCAACAAAAUCAAACCA GUUUGAUUUUGUUGAAAAUCC	−18.34, 35.22
11	14,005–14,027	acgcaaattgcgttaattgtact	UACAAUUAACGCAAUUUGCGU GCAAAUUGCGUUAAUUGUACU	−22.34, 79.46
12	14,389–14,411	tggtatctaaaggtttctttaag	UAAAGAAACCUUUAGAUACCA GUAUCUAAAGGUUUCUUUAAG	−22.04, 67.95
13	16,177–16,199	gtcttgtattcggcttatacaag	UGUAUAAGCCGAAUACAAGAC CUUGUAUUCGGCUUAUACAAG	−26.53, 58.68
14	16,217–16,239	tccttctatagttgaatttaata	UUAAAUUCAACUAUAGAAGGA CUUCUAUAGUUGAAUUUAAUA	−20.24, 48.81
15	17,283–17,305	gtctacaataataaattgttagc	UAACAAUUUAUUAUUGUAGAC CUACAAUAAUAAAUUGUUAGC	−17.87, 75.42
16	17,583–17,605	aacaacattaacagatttaatgt	AUUAAAUCUGUUAAUGUUGUU CAACAUUAACAGAUUUAAUGU	−19.59, 62.23
17	18,028–18,050	ctctacaattaggattttcaact	UUGAAAAUCCUAAUUGUAGAG CUACAAUUAGGAUUUUCAACU	−22.08, 53.94
18	19,806–19,828	ttgtataagaaagtcaataatga	AUUAUUGACUUUCUUAUACAA GUAUAAGAAAGUCAAUAAUGA	−19.97, 64.53
19	20,090–20,112	ctcaactattcataactatttta	AAAUAGUUAUGAAUAGUUGAG CAACUAUUCAUAACUAUUUUA	−19.63, 42.01
20	20,498–20,520	tgccaatatgcgtgttatacatt	UGUAUAACACGCAUAUUGGCA CCAAUAUGCGUGUUAUACAUU	−25.98, 74.21
21	20,948–20,970	gggtactattaaagaaaatatag	AUAUUUUCUUUAAUAGUACCC GUACUAUUAAAGAAAAUAUAG	−17.65, 66.76

S.N.: serial number

**Table 2 molecules-26-02610-t002:** Cytotoxicity 50% (CC_50_) and inhibitory 50% (IC_50_) concentrations of the different siRNAs.

siRNAs	CC_50_ (nM)	IC_50_ (nM)	
		Supernatant	Lysate
siRNA-1	>100	ND	ND
siRNA-2	>100	ND	ND
siRNA-3	>100	ND	ND
siRNA-4	>100	ND	ND
siRNA-5	>100	ND	ND
siRNA-6	>100	26.22	17.56
siRNA-7	>100	ND	ND
siRNA-8	>100	ND	ND
siRNA-9	>100	ND	ND
siRNA-10	>100	ND	ND
siRNA-11	>100	ND	ND
siRNA-12	>100	ND	ND
siRNA-13	>100	ND	ND
siRNA-14	>100	ND	ND
siRNA-15	>100	ND	ND
siRNA-16	>100	13.87	16.25
siRNA-17	>100	ND	ND
siRNA-18	>100	ND	ND
siRNA-19	>100	11.12	5.17
siRNA-20	>100	ND	ND
siRNA-21	>100	ND	ND

## Data Availability

Data is contained within the article.

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
