# Peer review of "In Vitro Inhibitory Analysis of Rationally Designed siRNAs against MERS-CoV Replication in Huh7 Cells"

_molecules, 2021, doi:10.3390/molecules26092610_

Round 1
Reviewer 1 Report
In this article Sherif Aly El-Kafrawy et al., use bioinformatics and in vitro cell culture system for siRNAs testing towards MERS-CoV replication. The article seems more like a proof-of-concept rather than a true identification of potential molecules for future clinic use. Even more 10 of the siRNA sequences were previously published and tested in a cell-culture system, so I find difficult to uncover the novel insights provided by the current manuscript.
Major points:
- Are there any particular reasons for the authors choosing to work in Huh7 cells? These are hepatocarcinoma cells, which are already modified by the tumor stress and their approach would have been more feasible in a normal cell line, maybe one more appropriate for the entry points of the virus.
- It is worth mentioning that the authors have previously performed in silico prediction and wet lab testing experiments for 10 of the described siRNAs in a different cell culture system. Although this is mentioned in the Discussion paragraph I believe this should be clearly described from the beginning of the manuscript by pointing the reader to the corresponding reference. Even more, the authors should amend the Results section (P3-4, L105-123) of the manuscript in which they mention that they found 21 functional siRNAs and indicate that actually 10 of the sequences were previously published and described (PMID: 33493920) and only 4 showed a response towards virus replication inhibition. This is important for the context of the study to underline only the new findings in the manuscript.
- P5-6, L124-137: The authors do no make any comments regarding the differences between the secondary structure of each studied RNA: "The secondary structure of each siRNAs and centroid secondary structure of ORF1ab region are presented in Figure 2a and Figure 2b." They should stipulate what information these secondary structures provide in the context their potential use. Just simply providing a note regarding the figure does not help the reader to easily understand the differences between these. The same observation is valid for the accessibility plots. For example there is a difference between 20 and the other plots. What does this indicate? Also adding labels to both axes would help in interpreting the plot much easier.
- How does secondary structure prediction help to filter only for best potential hits regarding inhibition efficiency? Could some correlations between the secondary structure, accessibility and their EC50 profile be made (2,16,6 and 19), compared with the other sequences?
- Table 2: The authors show a high concentration regarding the IC50 value for siRNA-2, greater than the CC50 calculated value (100 nM) and much higher compared with the other three siRNAs tested. They should explain this discrepancy in the Results section of the manuscript.
- Figure 4a and b: Inhibition curves are calculated only from four points. These should have at least 8 points for a better accuracy. Although the authors state under the Materials and Methods section that the experiments were performed in triplicates, no error bars are shown in the graphical representation. Thus the figures should have SD (standard deviation) error bars.
Minor points:
- P1, abstract, L14: Subse-quently probably should be subsequently.
- P1, abstract, L18: Please delete repetition: A total of total… should be A total of…
- P1, L35: ‘…illness in 60 years old patient…’ should be ‘illness in a 60 years old patient’
- P1, L40: CFR… probably is Case Fatality Rate (CFR). All abbreviations should be described.
- There is a discrepancy for the references. These are numbered in the manuscript text. However in the Reference section I cannot find the numbers, these are probably in an alphabetical order. This should be corrected.
- P3, Figure 1: Legend in Figure 1 should be more descriptive. What does represent exactly each sequence? Why the alignment is showing only 1:100 and 29910:30000? What does the boxed regions represent?
- P3, L115: “…as this will influence the siRNA influences…” Please delete the repetition.
- Figure 4a and b: Discrepancy between the title and legend for each of the two figures. Figure 4a title suggests that the EC50 plotted values are from the cell lysate, but the figure legend indicates that the values are from supernatant. Similar for Figure 4b.
- P9, L217: Probably missing word: “…cells and found that siRNAs 1 and 4 were most effective in Vero cells while siRNAs 1, 2, 4, 6 and 9 were most effective.”
- Figure 2: What do the colors in Figure 2a denote? Please add a graphical legend or describe in the figure legend text.
- The legend in Table 2 is confusing: “The average OD value of Cytotoxicity of different siRNAs at various concentrations.”, but the table describes both IC50 and CC50 values. I would suggest splitting the results from Table 2 or reordering the information: one table should contain only the cytotoxicity results (CC50) or simply state that the same value was obtained for all tested siRNAs and the other table the results from real time PCR with the IC50 values. This is because the text from paragraph 2.4 refers to Table 2, but the reader finds first the IC50 values not CC50 and this creates confusion. Also, paragraph 2.5 refers to the IC50 results, but the table is not mentioned in the paragraph.
Author Response
|
|
Yes |
Can be improved |
Must be improved |
Not applicable |
|
|
Does the introduction provide sufficient background and include all relevant references? |
( ) |
(x) Now improved |
( ) |
( ) |
|
|
Is the research design appropriate? |
( ) |
(x) Now improved |
( ) |
( ) |
|
|
Are the methods adequately described? |
( ) |
( ) |
(x) Now improved |
( ) |
|
|
Are the results clearly presented? |
( ) |
( ) |
(x) Now improved |
( ) |
|
|
Are the conclusions supported by the results? |
( ) |
( ) |
(x) Now improved |
( ) |
|
Major points:
- Are there any particular reasons for the authors choosing to work in Huh7 cells? These are hepatocarcinoma cells, which are already modified by the tumor stress and their approach would have been more feasible in a normal cell line, maybe one more appropriate for the entry points of the virus.
Response: The Huh 7 cells were used based on their better growth and better replicative capacity of MERS-CoV in cell lines (Eckerle et al. Replicative Capacity of MERS Coronavirus in Livestock Cell Lines. Emerg Infect Dis. 2014;20(2):276-279. doi:10.3201/eid2002.131182) and they have been used in many antiviral studies. In our previous report, we have evaluated some of the designed siRNAs in VERO cells (REF) which is an animal cell line while in this study we have chosen Huh 7 cell lines as a mean of evaluating the siRNA in a human cell line as a pre-step to use them in humans if proven effective. This statement was added to the end of the introduction part as “Huh 7 cell lines were selected as a proof of concept to evaluate the cytotoxicity and antiviral activity of siRNA in a human cell line in order to show their safety and efficacy for future use in preclinical and clinical trials.”
- It is worth mentioning that the authors have previously performed in silico prediction and wet lab testing experiments for 10 of the described siRNAs in a different cell culture system. Although this is mentioned in the Discussion paragraph I believe this should be clearly described from the beginning of the manuscript by pointing the reader to the corresponding reference. Even more, the authors should amend the Results section (P3-4, L105-123) of the manuscript in which they mention that they found 21 functional siRNAs and indicate that actually 10 of the sequences were previously published and described (PMID: 33493920) and only 4 showed a response towards virus replication inhibition. This is important for the context of the study to underline only the new findings in the manuscript.
Response: We Thank the reviewer for this comment. As per reviewer’s comments and suggestions, we have added this statement at the end of the introduction part “We have previously performed in silico prediction and a pilot study for investigating the antiviral activity for 10 of the described siRNAs in a different cell culture system [40]”. We also have amended the Results section as suggested.
- P5-6, L124-137: The authors do not make any comments regarding the differences between the secondary structure of each studied RNA: "The secondary structure of each siRNAs and centroid secondary structure of ORF1ab region are presented in Figure 2a and Figure 2b." They should stipulate what information these secondary structures provide in the context their potential use. Just simply providing a note regarding the figure does not help the reader to easily understand the differences between these. The same observation is valid for the accessibility plots. For example there is a difference between 20 and the other plots. What does this indicate? Also adding labels to both axes would help in interpreting the plot much easier.
Response: As per suggestions and comments, we have provided sufficient information and edited the text in our revised MS about secondary structure, thermodynamics accessibility plot as well as centroid secondary structure and their importance in binding capacity with the target.
- How does secondary structure prediction help to filter only for best potential hits regarding inhibition efficiency? Could some correlations between the secondary structure, accessibility and their EC50 profile be made (2,16,6 and 19), compared with the other sequences?
Response: Yes, the secondary structure of viral RNA is important regulator of virus biology because it plays an important role in virion ribonucleoproteins and virus assembly and release of infectious virus particles. The asymmetry of siRNA duplex plays a significant role to facilitate the assembly of the RNA-induced silencing complex (RISC). The estimation of target accessibility and duplex asymmetry can improve the target knockdown level significantly by nearly 40% and 26%, respectively. The duplex asymmetry has significant upstream effect on RISC assembly and target accessibility has strong downstream effect on target recognition
- Table 2: The authors show a high concentration regarding the IC50 value for siRNA-2, greater than the CC50 calculated value (100 nM) and much higher compared with the other three siRNAs tested. They should explain this discrepancy in the Results section of the manuscript.
Response: We apologize for this error, siRNA-2 was one of the siRNAs that had no or very low activity and was removed from the figure and text in the revised version.
- Figure 4a and b: Inhibition curves are calculated only from four points. These should have at least 8 points for a better accuracy. Although the authors state under the Materials and Methods section that the experiments were performed in triplicates, no error bars are shown in the graphical representation. Thus the figures should have SD (standard deviation) error bars.
Response: We apologize for this technical error, but something went wrong with Graphpad Prism that it was showing only four points. We performed the experiments in triplicate wells then they were pooled together to give enough volume for the extraction this is why there is no error bars on the figures. We have added the following statement in the experimental section, paragraph 4.7 “All the experiments were performed in triplicates, the triplicate well of each concentration were pooled together to provide enough volume for subsequent RNA extraction.”
Minor points:
- P1, abstract, L14: Subse-quently probably should be subsequently.
Response: the word was corrected to “subsequently” in the revised version.
- P1, abstract, L18: Please delete repetition: A total of total… should be A total of…
Response: the repetition was removed.
- P1, L35: ‘…illness in 60 years old patient…’ should be ‘illness in a 60 years old patient’
Response: the statement was changed to “a 60 years old patient” in the revised version.
- P1, L40: CFR… probably is Case Fatality Rate (CFR). All abbreviations should be described.
Response: the full words of the abbreviation was changed to” Case Fatality Rate (CFR)” in the revised version.
- There is a discrepancy for the references. These are numbered in the manuscript text. However in the Reference section I cannot find the numbers, these are probably in an alphabetical order. This should be corrected.
Response: We have now used the numbers for the list of references.
- P3, Figure 1: Legend in Figure 1 should be more descriptive. What does represent exactly each sequence? Why the alignment is showing only 1:100 and 29910:30000? What does the boxed regions represent?
Response: This is the Multiple sequences alignment result; we have selected only the start and end part of the genome to show the homology between the sequences due to the large size of virus genome and the image and space restriction in the MS, we have shown only selected regions as image.
- P3, L115: “…as this will influence the siRNA influences…” Please delete the repetition.
Response: As suggested, we have deleted the repetitive word.
- Figure 4a and b: Discrepancy between the title and legend for each of the two figures. Figure 4a title suggests that the EC50 plotted values are from the cell lysate, but the figure legend indicates that the values are from supernatant. Similar for Figure 4b.
Response: We apologize for this typing error, we have corrected the figure legends.
- P9, L217: Probably missing word: “…cells and found that siRNAs 1 and 4 were most effective in Vero cells while siRNAs 1, 2, 4, 6 and 9 were most effective.”
Response: We apologize for this typing error, the statement was corrected as follows “cells and found that siRNAs 1 and 4 were the most effective in Vero cells while siRNAs 1, 2, 4, 6 and 9 were most effective in HEK 293 cells” in the revised version.
- Figure 2: What do the colors in Figure 2a denote? Please add a graphical legend or describe in the figure legend text.
Response: In figure 2a, the color indicates base color and was added to the figure legend as “The colors indicate base type, blue: U, orange: G, light green: A, Light blue: U.”. While in figure 2b, the color is indicative of the secondary structure as follows; Green: Stems (canonical helices), Red: Multiloops (junctions), Yellow: Interior Loops, Blue: Hairpin loops. This image was generated by online software (RNA fold).
- The legend in Table 2 is confusing: “The average OD value of Cytotoxicity of different siRNAs at various concentrations.”, but the table describes both IC50 and CC50 values. I would suggest splitting the results from Table 2 or reordering the information: one table should contain only the cytotoxicity results (CC50) or simply state that the same value was obtained for all tested siRNAs and the other table the results from real time PCR with the IC50 values. This is because the text from paragraph 2.4 refers to Table 2, but the reader finds first the IC50 values not CC50 and this creates confusion. Also, paragraph 2.5 refers to the IC50 results, but the table is not mentioned in the paragraph.
Response: We apologize for this confusion, table 2 was modified to include the CC50 values first followed by the IC50 values for the siRNA. And the table was mentioned in paragraph 2.5.
Reviewer 2 Report
March 27, 2021
Review on “In vitro inhibitory analysis of rationally designed siRNAs against MERS-CoV replication in Huh7 Cells” by Sherif Aly El-Kafrawy, Sayed Sartaj Sohrab, Zeenat Mirza, Ahmed M. Hassan, Fatima Alsaqaf and Esam Ibraheem Azhar
The authors report the results of a rational design of siRNAs against MERS-CoV replication in Huh7 Cells. They used online software for designing the putative siRNAs, and subsequently performed the transfection and cytotoxicity assay as well as the evaluation of virus replication inhibition using real-time PCR. They noticed that no cytotoxicity was observed for any siRNAs in Huh7 cells and these siRNAs inhibit the viral replication in both cell lysate and supernatant. Finally, they conclude that the prediction of siRNAs using online software resulted in the filtration of potential siRNAs with high accuracy and strength. Considering that their in-silico design was followed by the experimental validation in a cell line, their conclusion seems valid. However, the data was poorly presented, and no thorough analysis (particularly on the in-silico data) was done (or presented) for the siRNA candidates. With the current form of the manuscript, I am not able to judge the significance of their work and suitability for publication. I strongly suggest authors to reconsider the shape and data presentation of the manuscript such that the main point of the work is clearly seen.
Some comments are followed:
- I suggest the authors to clarify the main point of the work: is the design protocol main or the designed siRNAs main? In the current form of the manuscript, neither of them was focused. For example, if the protocol is main, the author could provide a schematic illustration of a series of processes and give more detail analysis and description for each step of design.
- The design procedure in detailed should be provided either in the main text of SI. The statement like “By using stringent strategy,” (in P3) does not signify anything.
- Figures must be presented with sufficiently detailed captions.
- Figures 2-4 are very difficult to see and understand: no labels, no ticks, no units, etc. These figures do not help the understanding of design procedure on their own, and even they were almost not discussed in the main text.
- Table 2 should be compacted.
- References should be provided with the numbers.
Author Response
|
Yes |
Can be improved |
Must be improved |
Not applicable |
||
|
Does the introduction provide sufficient background and include all relevant references? |
( ) |
(x) Now improved |
( ) |
( ) |
|
|
Is the research design appropriate? |
( ) |
(x) Now improved |
( ) |
( ) |
|
|
Are the methods adequately described? |
( ) |
( ) |
(x) Now improved |
( ) |
|
|
Are the results clearly presented? |
( ) |
( ) |
(x) Now improved |
( ) |
|
|
Are the conclusions supported by the results? |
( ) |
( ) |
(x) Now improved |
( ) |
|
Comments and Suggestions for Authors
Review on “In vitro inhibitory analysis of rationally designed siRNAs against MERS-CoV replication in Huh7 Cells” by Sherif Aly El-Kafrawy, Sayed Sartaj Sohrab, Zeenat Mirza, Ahmed M. Hassan, Fatima Alsaqaf and Esam Ibraheem Azhar
The authors report the results of a rational design of siRNAs against MERS-CoV replication in Huh7 Cells. They used online software for designing the putative siRNAs, and subsequently performed the transfection and cytotoxicity assay as well as the evaluation of virus replication inhibition using real-time PCR. They noticed that no cytotoxicity was observed for any siRNAs in Huh7 cells and these siRNAs inhibit the viral replication in both cell lysate and supernatant. Finally, they conclude that the prediction of siRNAs using online software resulted in the filtration of potential siRNAs with high accuracy and strength. Considering that their in-silico design was followed by the experimental validation in a cell line, their conclusion seems valid. However, the data was poorly presented, and no thorough analysis (particularly on the in-silico data) was done (or presented) for the siRNA candidates. With the current form of the manuscript, I am not able to judge the significance of their work and suitability for publication. I strongly suggest authors to reconsider the shape and data presentation of the manuscript such that the main point of the work is clearly seen.
Response: We thank the reviewer for this comment, we have revised the MS with better presentation of data and provided the detailed information about the bioinformatics work.
Some comments are followed:
- I suggest the authors to clarify the main point of the work: is the design protocol main or the designed siRNAs main? In the current form of the manuscript, neither of them was focused. For example, if the protocol is main, the author could provide a schematic illustration of a series of processes and give more detail analysis and description for each step of design.
Response: The main objective of this work is to evaluate the antiviral efficacy of the designed siRNAs.
- The design procedure in detailed should be provided either in the main text of SI. The statement like “By using stringent strategy,” (in P3) does not signify anything.
Response: As suggested, we have corrected and edited the text in our revised MS. We have already published the designing protocol as mentioned above. In previous publications, we have reported on the protocols and procedure for in silico design and these publications were cited in paragraph 4.2 of the materials and methods of the revised version.
- Figures must be presented with sufficiently detailed captions.
Response: We thank the reviewer for this comment, we have provided a more detailed description of the figures in the figure captions.
- Figures 2-4 are very difficult to see and understand: no labels, no ticks, no units, etc. These figures do not help the understanding of design procedure on their own, and even they were almost not discussed in the main text.
Response: We thank the reviewer for this comment, we have provided the detail about the figures in text as well as captions.
- Table 2 should be compacted.
Response: As suggested, table 2 has been revised and updated.
- References should be provided with the numbers.
Response: We have now provided the numbers for the references.
Round 2
Reviewer 2 Report
April 14, 2021
Review on “In vitro inhibitory analysis of rationally designed siRNAs against MERS-CoV replication in Huh7 Cells” by Sherif Aly El-Kafrawy, Sayed Sartaj Sohrab, Zeenat Mirza, Ahmed M. Hassan, Fatima Alsaqaf and Esam Ibraheem Azhar
I find some improvements in the revised MS, but unfortunately, the data was still poorly presented with no thorough analysis. The authors improved Figure 2-4 in visibility and caption but no through analysis followed. The main focus is to evaluate the antiviral efficacy of the designed siRNAs as the authors stated, while it largely relies on the in-silico prediction and design. The authors state in the revised MS the importance of secondary structure information and target accessibility. However, no discussion on their own data can be found. What are the features of secondary structures found in the selected species? What is the message in Figure 2b? How should readers look at a series of plots in Figure 3 (Figure resolution is too low and cannot see the labels in Fig. 3, so I might be misunderstood the meaning of this plots). The authors and readers could benefit from such analysis results. The manuscript might be publishable, but the authors should provide the results of analysis on their in-silico data.
Author Response
Response to comments
(x) I would not like to sign my review report
( ) I would like to sign my review report
English language and style
( ) Extensive editing of English language and style required
( ) Moderate English changes required
(x) English language and style are fine/minor spell check required
( ) I don't feel qualified to judge about the English language and style
|
Yes |
Can be improved |
Must be improved |
Not applicable |
|
|
Does the introduction provide sufficient background and include all relevant references? |
(x) |
( ) |
( ) |
( ) |
|
Is the research design appropriate? |
(x) |
( ) |
( ) |
( ) |
|
Are the methods adequately described? |
( ) |
(x)Improved now |
( ) |
( ) |
|
Are the results clearly presented? |
( ) |
( ) |
(x) Improved now |
( ) |
|
Are the conclusions supported by the results? |
( ) |
( ) |
(x) Improved now |
( ) |
Comments and Suggestions for Authors
April 14, 2021
Review on “In vitro inhibitory analysis of rationally designed siRNAs against MERS-CoV replication in Huh7 Cells” by Sherif Aly El-Kafrawy, Sayed Sartaj Sohrab, Zeenat Mirza, Ahmed M. Hassan, Fatima Alsaqaf and Esam Ibraheem Azhar
I find some improvements in the revised MS, but unfortunately, the data was still poorly presented with no thorough analysis. The authors improved Figure 2-4 in visibility and caption but no through analysis followed. The main focus is to evaluate the antiviral efficacy of the designed siRNAs as the authors stated, while it largely relies on the in-silico prediction and design. The authors state in the revised MS the importance of secondary structure information and target accessibility. However, no discussion on their own data can be found. What are the features of secondary structures found in the selected species? What is the message in Figure 2b? How should readers look at a series of plots in Figure 3 (Figure resolution is too low and cannot see the labels in Fig. 3, so I might be misunderstood the meaning of this plots). The authors and readers could benefit from such analysis results. The manuscript might be publishable, but the authors should provide the results of analysis on their in-silico data.
Submission Date
18 March 2021
Date of this review
14 Apr 2021 11:39:46
Response:
Thanks for the critical observation and comments and suggestions. Based on the suggestions, we have now edited our MS with discussion of our own data in the text. We have also discussed about the features of secondary structure found in the selected species. The message in figure 2b is the centroid secondary structure of MERS-orf1ab gene. Based on the suggestions, we have used new structure generated by online software. This is the best picture we were able to generate and downloaded from software with highest resolution. The figure showed each nucleotide position in the viral genome. Base on the nucleotide position, the location of each siRNAs can be identified in the genome. Figure 3 is the target accessibility plot of each siRNAs. The plot shows the exact binding position of siRNAs with high efficiency in a specified target with high accessibility. We have generated these figures by using online software and we were able to download the best picture with the highest resolution allowed by software.